# Intestinal Absorption Study: Challenges and Absorption Enhancement Strategies in Improving Oral Drug Delivery

**DOI:** 10.3390/ph15080975

**Published:** 2022-08-08

**Authors:** Maisarah Azman, Akmal H. Sabri, Qonita Kurnia Anjani, Mohd Faiz Mustaffa, Khuriah Abdul Hamid

**Affiliations:** 1Department of Pharmaceutics, Faculty of Pharmacy, Universiti Teknologi MARA Cawangan Selangor, Puncak Alam 42300, Selangor, Malaysia; 2Medical Biology Centre, School of Pharmacy, Queen’s University Belfast, 97 Lisburn Road, Belfast BT9 7BL, UK; 3Fakultas Farmasi, Universitas Megarezky, Jl. Antang Raya No. 43, Makassar 90234, Indonesia; 4Department of Pharmacology and Pharmaceutical Chemistry, Faculty of Pharmacy, Universiti Teknologi MARA Cawangan Selangor, Puncak Alam 42300, Selangor, Malaysia; 5Atta-ur-Rahman Institute for Natural Product Discovery (AuRINS), Universiti Teknologi MARA Cawangan Selangor, Puncak Alam 42300, Selangor, Malaysia

**Keywords:** absorption enhancers, intestinal absorption, oral delivery, pharmacokinetic profile, nanocarriers

## Abstract

The oral route is the most common and practical means of drug administration, particularly from a patient’s perspective. However, the pharmacokinetic profile of oral drugs depends on the rate of drug absorption through the intestinal wall before entering the systemic circulation. However, the enteric epithelium represents one of the major limiting steps for drug absorption, due to the presence of efflux transporters on the intestinal membrane, mucous layer, enzymatic degradation, and the existence of tight junctions along the intestinal linings. These challenges are more noticeable for hydrophilic drugs, high molecular weight drugs, and drugs that are substrates of the efflux transporters. Another challenge faced by oral drug delivery is the presence of first-pass hepatic metabolism that can result in reduced drug bioavailability. Over the years, a wide range of compounds have been investigated for their permeation-enhancing effect in order to circumvent these challenges. There is also a growing interest in developing nanocarrier-based formulation strategies to enhance the drug absorption. Therefore, this review aims to provide an overview of the challenges faced by oral drug delivery and selected strategies to enhance the oral drug absorption, including the application of absorption enhancers and nanocarrier-based formulations based on in vitro, in vivo, and in situ studies.

## 1. Introduction

Many studies have been conducted over the past few decades to identify the best permeation enhancers for poorly absorbed drugs that are administered orally. It is important to highlight this issue as patients have shown their preference for oral formulations over other dosage forms, as reported by a survey where 71.91% of the interviewed patients favored the daily intake of an oral formulation relative to those that preferred a daily subcutaneous injection, thus highlighting the need for improved oral drug delivery [1]. This leads to the discovery of a variety of permeation enhancers that have shown great potential in improving the delivery efficiency of orally administered drugs, as shown by an increase in the number of publications over the years.

The majority of drugs intended for systemic effects are administered orally, as the oral route is the most convenient, practical, and the preferred route for drug delivery. However, one of the major limitations of orally administered drugs is the need for the molecule to undergo intestinal absorption before it can be distributed to the intended target site where the drug can elicit its therapeutic effects [2]. The permeability of drugs across the intestinal membrane is one of the key factors that governs the pharmacokinetic profile of the drug molecules, with approximately 90% of drug absorption occurring in the small intestines [3]. 

According to the Biopharmaceutics Classification System (BCS) of the US Food and Drug Administration, the intrinsic properties of drugs that influence oral absorption are their aqueous solubility and the permeability of the drugs across the intestinal membrane [4]. Solubility is one of the properties that influences the oral bioavailability of drug molecules due to its role in drug dissolution within the gastric lumina [5]. However, high solubility alone is insufficient to provide the required bioavailability that is needed to elicit the desired pharmacological effects. The delivered drug also needs to possess a high membrane permeability, which governs how a dissolved drug molecule could be transported across the intestinal membrane before reaching the systemic circulation [2]. Based on the BCS classifications, the drugs classified in class III and class IV have a low membrane permeability [4], and therefore suffer the challenge of limited transport across the intestinal epithelial layer. Another key factor in the oral bioavailability parameters that should be focused on is the intestinal permeability rate, which connects the drug disposition to the degree of drug metabolism, as described by Wu and Benet (2005) in the Biopharmaceutics Drug Disposition Classification System (BDDCS) [6].

There appears to be a rapidly growing interest in the drug formulation to improve the delivery of orally administered drugs, which began in 1961 in the work of Wagner [7], who highlighted the fundamentals of drug absorption, the relationships between the dosage forms, the optimal dosage schedules, the kinetics of absorption, and the hypothetical models of the kinetics of absorption and excretion. This subsequently progressed to the study of the relative absorption of novobiocin–tetracycline combinations and tetracycline hydrochloride alone in capsules for children and adults [8], and a further detailed study on pharmacokinetics in 1972 which reported on the absorption, metabolism, and excretion of methylene blue in man and dog after oral administration [9]. These findings laid the foundation for oral-based pharmaceutical research, which numerous scientists have referred to and cited. 

This review highlights the challenges of oral drug absorption, which focuses on the environmental factors within and along the gastrointestinal tract. Due to these limitations, there is an impetus to enhance the absorption of poorly absorbable drugs via the use of permeation enhancers. Hence, this article also outlines the application of various compounds that have been utilized as permeation and absorption enhancers. The effectiveness of these compounds, when evaluated at a pre-clinical level used in vivo, in vitro, and in situ permeability studies, will be discussed. 

The first section of the review provides a general background on the challenges typically encountered when delivering a therapeutic compound through the oral route. These barriers include the diffusion of the compound across the gut wall, the back transport into the gastric lumina due to efflux transporter, along with issues related to the first-pass metabolism. The second section of the review places the emphasis on potential strategies to enhance the solubility of the poorly water-soluble drugs in order to enhance oral bioavailability. The third section of the review outlines the application of common nano-systems, such as solid lipid nanocarriers, nanoemulsions, and dendrimers as promising drug delivery platforms to deliver a high concentration of poorly absorbed drugs across the gastrointestinal tract in order to enhance the oral bioavailability. The progress that has been made in utilizing nanocarriers for oral drug delivery will be discussed, based on in vivo, in vitro, and in situ permeability studies. It is hoped that this review will provide an introduction for formulation scientists and new researchers in the area of oral drug delivery on the approaches and strategies for enhancing solubility and promoting intestinal absorption of drug molecules across the gastrointestinal tract, with the ultimate aim of improving oral bioavailability and systemic efficacy. 

## 2. The Challenges in Oral Drug Delivery

The gastrointestinal tract is mainly involved in the digestion and absorption of nutrients, along with the excretion of waste products. As a pharmaceutical formulation enters the mouth, it will undergo mastication and partial digestion to facilitate its movement as a bolus from the esophagus into the stomach. The stomach then takes over the digestion process with the aid of gastric acid and enzymes, such as pepsin and lipase [10]. Consequently, the formulation enters the small intestine, where the active drug molecule is released and its absorption occurs through simple diffusion, facilitated diffusion, or even secondary active transport [11]. 

The enhanced absorption of oral drugs within the small intestine may be facilitated by the presence of multiple digestive enzymes, such as lipase and peptidase, along with the exocrine secretions of zymogen, such as trypsinogen, chymotrypsinogen, and procarboxypeptidase, which are released from the biliary system and the pancreas [12]. The absorption of the drug molecules is also aided by the villi and microvilli, which increase the intestinal surface area by 30–600 fold [12]. However, the presence of barriers, such as mucous, tight junctions, efflux transporters, and enzymes, limits the absorption of certain drugs. The overview of these barriers will be discussed further to understand their influence on oral drug absorption. 

### 2.1. Mucous

The small intestine is the main site for drug absorption, as it has a high surface area of approximately 400 m^2^ [13]. The microvilli are lined with goblet cells whose major role is to secrete the glycoproteins that form the mucosal lining of the small intestine. The mucous is a rigid layer that is also composed of proteins, nucleic acids, mucins, and electrolytes, which mainly acts as a buffer by regulating the pH at six along the apical surface, thereby creating an acidic mantel that lines the small intestine [14]. This mucous covers the epithelial cells of the intestinal lumen and is bounded to the apical surface by the glycocalyx. Both the glycocalyx and the mucosal layer are involved in the formation of an unstirred water layer (UWL) that possesses a thickness of approximately 100 μm [15], which separates the brush border of the enterocytes from the bulk fluid phase of the small intestine lumen. Due to the low aqueous solubility of the lipophilic drugs, these molecules face difficulties in crossing the UWL to gain access to the brush border membrane where absorption can take place [16]. Examples of the therapeutics by which their absorption are affected by the presence of GI mucous include peptides and proteins in the range of 14.4–168 and 3.4–66 kDa [17], as well as small drug molecules such as albuterol, rifampicin, p-amino-salicylic acid, isoniazid, pyrazinamide, pentamidine, pivampicillin, isoprenaline (isoproterenol), isoetarine, and rimiterol [18].

The mucosal layer has different thicknesses and turnover values which are dependent on the pathophysiological status (e.g., gastritis and ulcer) of the gastrointestinal tract, interaction with the external environment, and the anatomical region [19]. The thickness of the mucus blanket affects the drug absorption in such a way that any irritants imposed upon the gastrointestinal tract stimulate the mucosal thickening, thus making it harder for the drug to penetrate this layer to reach the epithelium. It is a continuous cycle of mucous secretion and shedding from the gastrointestinal membrane surface; therefore, this layer becomes a major barrier to drug delivery because the drug that is delivered to mucosal surfaces must migrate into and through the mucous within a defined period to gain access to the epithelium to avoid being shed as well [17]. Other than that, the mucosal barrier is intrinsically lipophilic, which is attributed to the non-glycosylated protein domains [20], and, in some of the regions, is negatively charged, owing to the mucin proteoglycans [21], which can serve as a physical barrier to the absorption of certain drugs. To date, researchers have developed an in vitro, in vivo and ex vivo mucus model to evaluate the oral drugs’ diffusivity in the presence of mucin, natural mucus, artificial mucus, or mucus produced by specialized cells (HT-29, Calu-3, Caco-2) [22,23].

### 2.2. Tight Junction

The absorption of the hydrophilic drugs through the intestinal epithelial layer is governed by the paracellular pathway. The paracellular route involves an intercellular junction consisting of desmosomes, adherent junctions, and tight junctions [24]. The discovery of the integral membrane components of the tight junctions marks the breakthrough in the significance of tight junctions. Occludin was first identified by the group of Shoichiro Tsukita two decades ago [25], followed by the discovery of claudin-1 and -2 [26]. These tight junctions function as a distinction between the basolateral and apical sides of the epithelial surface, while serving a role as rate-limiting step for paracellular transport [27]. Some of its components that are associated with the barrier function are occludins, zonula occludens 1 (ZO1), ZO2, and claudins. Among the proteins, claudin is the most essential protein that contributes to the barrier function of the tight junctions as it is involved in the formation of ion-selective paracellular pores. These proteins are expressed more in the jejunum than the ileum, thus play a more pivotal role in the paracellular transport of drugs within that particular intestinal region [28].

There are two pathways involved in paracellular permeability across the tight junctions. The first one is the epithelial leak pathway which is controlled by occludin and myosin light-chain kinase (MLCK). This pathway is known to be size-dependent but not charge-dependent. Only the materials whose radius gyrations are larger than 10 μm are capable of being transported via this route [24]. The epithelial leak pathway permits the movement of large compounds, such as bacterial lipopolysaccharides and proteins. In contrast, the pore pathway, which consists of the claudin proteins, is both size- and charge-dependent; therefore, it allows the permeation of polar drugs and excludes molecules larger than 4 Å [24]. As for charge selectivity, claudin prefers cations better than anions in which the flux magnitude range is established to vary among the epithelia [29,30]. Nevertheless, due to the small percentage of the overall total surface area, the drug transport across the intestinal epithelium via the paracellular pathway is considered to be minimal [3]. The examples of drugs and therapeutics whose absorption occurs through the tight junctions include the peptide drugs, such as octreotide, the vasopressin analog-desmopressin, and the thyrotropin-releasing hormone [31]. In addition, the work by Tam et al. has shown that amphoteric drugs, such as acyclovir, amdinocillin, ganciclovir, piroxicam, and trovafloxacin, undergo extensive paracellular transport via the tight junctions [32].

### 2.3. Efflux Transporters

The efflux transporters are integral transmembrane proteins that span across the plasma membrane of epithelial cells and are involved in transporter-mediated drug transport [33]. The ATP-binding cassette (ABC) transporter superfamily consists of several pertinent plasma-membrane efflux pumps. These include: (a) P-glycoprotein (P-gp; gene symbol ABCB1); (b) multidrug resistance-associated protein 1 (MRP1; gene symbol ABCC1); as well as (c) breast cancer resistance protein (BCRP; gene symbol ABCG2, also known as mitoxantrone resistance protein) [34]. These efflux pumps are responsible for pumping endogenous and exogenous substances out of the cells [11]. The transporters use energy from ATP hydrolysis to efflux the compounds back into the intestinal lumen. Examples of the efflux transporters include the multidrug resistance-associated protein (MRP), the P-glycoprotein (P-gp), and the breast cancer resistance protein (BCRP). The induction of the efflux transporters may affect the rate of drug absorption. Therefore, it is important to identify the drugs that are substrates of these efflux pumps to avoid drug resistance. Some of the P-gp efflux pump substrates include cyclosporine A, digoxin, atorvastatin, and methotrexate [35], while scutellarin, calcein, and ganciclovir are known substrates to the multidrug resistance-associated protein 2 (MRP2) [33]. 

The different types of the transporter are expressed along the intestinal tract, which gives rise to site-specific drug absorption. For instance, the peptide-proteins (PEPT1) that are involved in the uptake of norfloxacin are mostly expressed along the ileum, which is the main site for the absorption of the antibiotic, relative to the colon [36]. In addition, the P-glycoproteins are most abundant along the ileum and the colon which is the main cause for the reduced drug absorption for a range of molecules, such as cyclosporin, tacrolimus, and paclitaxel [36,37]. The process of drug transport by these efflux transporters is to some extent governed by the concentration of the substrates. For example, Duan et al. (2013) discovered that the permeability of capsaicin across jejunum, ileum, and colonic membranes increases when the concentration of the compound increases from 100 to 200 µM. However, the permeability of capsaicin remained unchanged across all three of the intestinal segments when the concentration of the compound was increased beyond 200–500 µM. This suggests that, indeed, the transport of the substrate by these efflux transporters is indeed concentration-dependent, however such a process may become saturated at high drug concentrations [38]. In addition, the pioneering work by Mouly and Paine that involved the Western blot analysis of human GI mucosal scrapings revealed that the relative distribution of the P-glycoproteins’ expression (P-gp/villin-integrated optical density ratio) progressively increases from the proximal to the distal regions of the small intestine [39]. Such changes in the distribution of the efflux transporter may account for the site-specific absorption that typically occurs for certain drug molecules. In addition, this seminal study also showed that such distribution in the P-glycoprotein also showed a considerable inter-individual variability (1.5- to 3.0-fold), which may serve as a potential explanation for the considerable variability in the pharmacokinetic profiles typically observed with the oral administration of drugs. Furthering this, Bruyére and co-workers also quantified cytochrome P450/3A4 (CYP3A4), P-glycoprotein (P-gp, ABCB1), and the breast cancer resistance protein BCRP (ABCG2), throughout the human small intestine from 19 donors in an attempt to develop a scaling factor for predicting the clearance of drug molecules along the GI tract, which may serve as a foundation in developing a physiologically based pharmacokinetic (PBPK) model [40]. Their result echoed the initial findings by Mouly and Paine in showing that P-glycoproteins expression indeed increases from the proximal to the distal region of the small intestine. However, the researchers also discovered that the expression of cytochrome CYP3A4, an integral enzyme responsible for the oxidation of xenobiotics such as drug molecules, decreased from the upper to the lower small intestine. In addition, they also discovered that BCRP expression did not vary significantly with position, but varied greatly between individuals. By reflecting upon the results discovered by Mouly and Paine in tandem with the findings by Bruyère et al., it could be postulated that a decrease in the intestinal absorption of certain drug molecules along the GI tract may arise from a combination of metabolism and efflux transport, with metabolism being the main mechanism in the proximal region of the small intestine, which is then superseded by efflux transport of the drug in the distal region of the small intestine.

Based on this finding, it can be postulated that an increase in a drug concentration to a level that can lead to efflux transporter saturation may serve as a strategy to enhance the transport of compounds across the membranes. However, such a transport is concentration-dependent, and any increase beyond the optimum concentration will not lead to an increase in intestinal membrane permeability, but may lead to wastage as any unabsorbed drug will just be excreted along the gastrointestinal tract. In addition, one must also consider that in order for this strategy to be effective, a high concentration of the drug must be delivered within the gastric lumina for the transporter to be saturated. Such a high concentration of the drug within the gastric lumina may lead to localized irritation along the gastrointestinal tract, along with the potential for localized side-effects.

### 2.4. Enzymes

The enzymes represent the biochemical barrier to drug absorption as they play a major role in drug degradation, thus indirectly reducing the oral bioavailability of the drug absorbed. Pancreatic and mucosal cells are capable of secreting enzymes directly into the lumina of the digestive tract creating a tortuous pathway rich in degradative enzymes that drug molecules must endure before the molecules are absorbed into the systemic circulation. In addition, peptidases are also present along the brush border membrane of the epithelial cells, which have a high selectivity for the breakdown of both peptides and proteins, such as insulin and calcitonin [41]. Furthermore, the local microflora within the colon also produce various enzymes that contribute to the drug degradation [42]. B-glucuronidase, which is generated by *Escherichia coli*, is able to catalyze the hydrolysis of various drug molecules, such as estradiol-3-glucuronide and chloramphenicol glucuronide [43]. 

Proteins are an example of biomolecules that are very susceptible to proteolytic enzymes as they can be cleaved into oligopeptides and ultimately into amino acids. Even before reaching the small intestine, proteins and peptide therapeutics are already exposed to pepsin within the stomach. Nevertheless, the majority of the protein degradation and absorption occurs within the small intestine. This process specifically occurs along the brush border of the villi and involves the enzymes, endopeptidases, exopeptidases, chymotrypsin, and trypsin, that contribute to the sequential breakdown of the proteins into peptides and eventually into amino acids [44]. 

Although enzymatic degradation affects the drug bioavailability before being absorbed into the systemic circulation; this process in some instances also has an integral role on the activation of certain drugs before these molecules can exert any pharmacological effect. This enzyme-mediated activation of certain drug molecules has led to the discovery of prodrug formulations. These are pharmaceutical formulations that deliver compounds with additional chemical moiety through the stomach and small intestine, but require biotransformation through an enzymatic reaction along the gastrointestinal tract to remove these additional chemical structures to generate active metabolites. The prodrug strategy not only confers protection to the active drug but also endows the active drug with other several advantages, such as increased absorption, mitigation of side effects, as well as circumventing metabolic inactivation [45]. In the management of inflammatory bowel disease (IBD), the prodrug strategy has been utilized to deliver the active molecule to the right region of the gastrointestinal tract. These prodrugs are the precursors to the active molecules that exert anti-inflammatory, immunomodulating, and immunosuppressive effects. For instance, Balsalazide, a prodrug of 5-Aminosalicylic acid (5-ASA), is poorly absorbed in the stomach and small intestine, thus permitting it to be selectively activated to 5-ASA by the azo-reducing bacteria present in the colon and to exert its anti-inflammatory action along the colon in IBD [46]. Meanwhile, omeprazole, a prodrug of sulfonamide, utilizes the acidic stomach to be activated and covalently bind to the cysteine residues of the proton pump (i.e., H+/K+-ATPase) on the luminal side in the oxyntic mucosa of the stomach, in order to exert its anti-ulcer effect [46].

### 2.5. First-Pass Metabolism

Metabolism can be defined as a series of biochemical degradations or modifications to the parent molecule prior to excretion. The product of the metabolic biotransformation is called a metabolite, which either possesses different pharmacological effects to its parent compound or becomes easier to be excreted. The small intestine can play an important role in the first-pass metabolism of certain drugs due to the presence of gastrointestinal drug-metabolizing enzymes that limit the drug bioavailability following oral administration, despite complete absorption. The drug that passes through the gastrointestinal tract will reach the liver via the enterohepatic circulation [43]. As the liver is the primary site of drug metabolism, the drug will undergo extensive biotransformation, which significantly reduces its bioavailability prior to systemic circulation, which can subsequently result in a subtherapeutic effect. 

In the liver, the drug biotransformation consists of phase I and phase II reactions. Generally, phase I reactions are the preparative stages that introduce polar chemical moieties, such as the hydroxyl, sulfide, and carboxylic acid groups, for the subsequent phase II processes. These modifications are performed via hydrolysis, oxidation, and reduction, which are mediated by the enzymes, such as cytochrome P450 monooxygenase, oxidase, reductase, and hydrolase. However, the polar metabolites may be directly excreted without undergoing the phase II reactions. In addition, some of the compounds may undergo primary elimination via phase II reactions only. Some of the prodrugs are converted to active metabolites, as is the case with codeine, which is demethylated to the more potent morphine [47].

The cytochrome P450 (CYP450) enzymes are pivotal in the phase I reaction. Within the human liver and along the intestinal wall, CYP3A comprises 70% of the total CYPs in the intestine [48]. Among the CYP3A subfamilies, CYP3A4 is quantitatively the major sub-family of CYP450 and plays a major role in the metabolism of many clinically important drugs of diverse chemical structures and sizes [49]. This causes the substrates of CYP3A4 to have a low bioavailability before they can reach systemic circulation and ultimately the target site of action. Some of the CYP3A4 substrates are midazolam, buspirone, verapamil, lovastatin, and cyclosporine. The sedative drug midazolam serves as a marker substrate to assess CYP3A4 activity that is widely present in the hepatic and intestinal microsomal tissue. One of the phase I metabolic reactions is oxidation, which involves the addition of an oxygen atom into the compound to form a polar hydroxyl group. Examples of drugs that are susceptible to oxidation are lignocaine, pentobarbitone, diazepam, and thiopurines. In some cases, oxidation is critical for the transformation of certain drugs into their more pharmacologically active form. For example, the O-dealkylation of codeine is involved in the formation of morphine, which displays greater analgesic properties [47]. This transforms the lipophilic drugs into more hydrophilic compounds that are easily excreted from the body. The prolonged exposure to a certain drug over a defined period may, in some instances, increase the metabolism of the drug and other compounds. This phenomenon is known as enzyme induction, and many drugs have been discovered as inducing the activity of most of the CYP450 enzymes.

The product of phase I reactions then undergoes conjugation with polar molecules (e.g., sulphate and glucuronic acid) in phase II reactions to form excretable hydrophilic metabolites. Some of the enzymes that mediate this process are glucuronosyltransferase, sulfotransferase, and N-acetyltransferase. An example of a phase II reaction is glucuronidation, which is catalyzed by uridine 5′-diphospho(UDP)-glucuronosyltransferase (UDPGT) which transforms morphine into the more potent morphine 6-glucuronide [47]. Additionally, ethinyl estradiol, terbutaline, and isoproterenol undergo sulfation via sulfotransferases [49].

### 2.6. Intestinal Lymphatic Transport

The intestinal lymphatic system is an alternative route for delivering highly lipophilic drugs to the body circulation. The drugs that have log octanol/water partition coefficients (log P) > 5 and a long-chain triglyceride (TG) solubility > 50 mg/g are unable to diffuse across the blood capillaries and will instead utilize the more permeable lymphatic capillaries for absorption [50,51]. During intestinal lymphatic drug transport, highly lipophilic drugs are prone to incorporation into chylomicrons in enterocytes. These chylomicron-associated drugs are then secreted into the mesenteric lymphatic circulation and rained directly into the systemic circulation at the intersection of the left jugular and subclavian veins. This route can bypass the liver, thus avoiding hepatic first-pass metabolism, which in turn increases oral bioavailability. Chemotherapeutic drugs, immunosuppressants, and anti-human immunodeficiency virus (anti-HIV) drugs enhance their therapeutic effects through this lymphatic absorption pathway [52]. 

Enterohepatic recycling is a phenomenon involving biliary excretion and the intestinal reabsorption of a solute, occasionally with hepatic conjugation and intestinal deconjugation. As the compound is absorbed into the portal venous blood by the enterocytes, it will be cleared from the systemic circulation by hepatic uptake and further secreted into the bile. Subsequently, these molecules are returned back into the intestinal lumen only to be reabsorbed by the intestinal cells and becoming available for enterohepatic cycling. Due to this cycling, the drugs that are susceptible and transported by this route may have a prolonged half-life, which results in extended pharmacological effects, such as furosemide and estradiol [43]. 

## 3. Current Absorption Enhancers and Their Absorption-Enhancing Mechanisms to Improve the Pharmacokinetic Profile

### 3.1. Solubilizing Agents

The application of solubilizing agents is one of the most common strategies to enhance the water solubility of orally administered drugs in an attempt to enhance the oral bioavailability of the delivered therapeutic compound. Solubilizing agents lead to the formation of fine dispersions of the lipid-solubilized drugs in the aqueous milieu of the gastrointestinal tract through the process of self-emulsification [53]. One of the classes of solubilizing agents are surfactants, which display amphiphilic properties. The surfactants are generally incorporated into emulsions and suspensions, due to their ability to reduce interfacial surface tension and leading to the production of a stable colloidal formulation. Its role as an absorption enhancer is contributed by its interaction with the plasma membrane, either by disrupting the barrier function of the epithelial cells or the sequestration of proteins [54]. In general, non-ionic surfactants are more favorable than ionic surfactant, as this sub-class of surfactants displays better safety profiles while only inducing a reversible modulation in the intestinal mucosal permeability [55]. 

One of the surfactants that have been studied as an absorption enhancer is sucrose laurate. Aside from its non-ionic characteristics, it has a low critical micelle concentration (CMC), which allows it to reduce the interfacial surface tension at a low concentration [56]. This makes it more potent than sodium caprate (C10) and sodium salt of lauric acid (C12), which have a much higher CMC thus necessitating a higher concentration of the surfactant in order to elicit a similar effect. In a study by McCartney et al., the group observed an increase in the P_app_ of [^14^C]-mannitol in the presence of sucrose laureate in a concentration-dependent manner in both in vitro and ex vivo studies [57]. This is supported by another study by Maher et al. that also reported an improvement in the mannitol permeability in the presence of sucrose laurate across isolated rat colonic mucosa [56]. Based on an in vivo study, sucrose laurate successfully improved the delivery of insulin at concentrations of 50 and 100 mM across rat jejunum and colon, resulting in the reduction in the blood glucose level. The absorption-enhancing action of this surfactant has been observed to affect the transcellular pathway. This result is associated with the ability of sucrose laurate to decrease the mitochondrial membrane potential while simultaneously increasing plasma membrane potential as well as altering the expression of the tight junctions protein, ZO-1 [57]. In addition, the surfactant is capable of fluidizing the plasma membrane while altering the efflux transporters’ activity that culminates in an increase in the substrate permeability across the membrane. Such an effect is site-dependent, with sucrose laurate having a more pronounced effect along the colon than the jejunum [57]. This effect on the membrane permeability by sucrose laurate along this region is also postulated to act in synergy with the high exposure of bile salts within the jejunum, which could improve drug solubility in the lumen. Aside from that, the lower peptidase level and longer residence time in the colon may, in tandem, reduce the drug degradation and improve the degree of absorption [58]. 

Another solubilizing agent that is frequently employed in drug delivery is cyclodextrins (CD). CDs are cyclic oligosaccharides with a hydrophobic center that enables the entrapment of drug molecules, as illustrated in Figure 1. Through encapsulating the hydrophobic drugs’ molecules inside the annulus, the cyclodextrins form an inclusion complex with the drug. This complex formation improves the drug solubility and aids the diffusion of the drug molecules across the unstirred water layer to reach the apical membrane of the enterocytes. In addition, this inclusion complex also improves drug permeability across the enterocytes, leading to improved oral bioavailability and pharmacokinetic profile [59].

The permeation-enhancing ability of cyclodextrin is also attributed to the ability of the cyclic oligosaccharide to extract the membrane proteins and phospholipids from the apical membrane, leading to enhanced membrane fluidization [60]. In addition, the formation of the cyclodextrin drug complex is also capable of widening the tight junctions along the gastrointestinal tract, leading to enhanced paracellular transport [61]. The formation of an inclusion complex allows the ‘guest’ molecule to be transported in the aqueous milieu of the gastrointestinal lumen, as the complex formation enhances the wettability, solubility, and dissolution rate of the drug while simultaneously improving drug stability, and permeability across the gastric mucosa [62]. Subsequently, this allows the inclusion complex to be transported across the lipophilic cell membranes. The CDs managed to enhance the oral absorption of BCS Class II drugs, such as carbamazepine and albendazole, which are characterized by low aqueous solubility, thus leading to improved oral bioavailability [63]. In a study by Rubim et al. (2017), such enhanced oral absorption and the bioavailability of the delivered amiodarone hydrochloride was attributed to the increase in the drug solubility when formulated with methyl-β-CD [64]. In another study by Devasari et al. (2015), erlotinib (ERL) was formulated with sulfobutyl ether beta-cyclodextrin, leading to the formation of drug-CD complex (ERL-SBE-β-CD) that enhanced the drug solubility by 7.4-fold [59]. In comparison to the free drug, the complex also showed a 5.4-fold decrease in T_max_, with a 3.2-fold increase in C_max_ when evaluated in vivo in Sprague Dawley rats, which was attributed to the enhanced dissolution of the drug. The studies that evaluated and elucidated the effect of surfactants and cyclodextrins on oral drug absorption are summarized in Table 1.

### 3.2. Bile Salts 

Bile salts are endogenous compounds that are present in the small intestine, with concentrations ranging from 8 mM and 18 mM in the fasted and fed state, respectively [66]. These compounds consist of taurine and glycine conjugates of chenodeoxycholic acid and cholic acid. The bile salts are categorized based on the extent of their hydroxylation and conjugation with amino acids. The classifications consist of dihydroxy conjugates (taurodeoxycholate and taurochenodeoxycholate), trihydroxy conjugates (glycocholate and taurocholate), and unconjugated forms (cholate and chenodeoxycholate) [67]. About 60% of the bile salts present within the gastrointestinal tract are dihydroxy conjugates. Through micellar solubilization, the bile salts play a major role in facilitating the digestion of dietary lipids via emulsification, lipolysis, and ultimately the transportation of the lipid products across the gut epithelium during absorption [14]. Should the bile salts escape active reabsorption within the ileum, the intestinal flora will metabolize them into secondary bile salts; lithocholic acid and deoxycholic acid. From a drug delivery perspective, the bile salts are capable of enhancing gastrointestinal absorption and ultimately oral bioavailability through several mechanisms, which include-solubilization of the poorly soluble drug through micellar solubilization, enhancing the chemical stability of the delivered therapeutic, increasing the membrane fluidity along the epithelial cells that line the gastrointestinal tract, the opening of tight junctions, membranolysis of the epithelial membrane, as well as modulating the function of the transport proteins along the gut epithelium [68,69]. Another mechanism for the absorption-enhancing effect of bile salts is the ability of the compound to reduce the viscosity and elasticity of the mucus layer, which in turn enhances the diffusion rate of the molecule across the mucus layer to the gut epithelium for intestinal absorption [69]. Another study also found that the bile salts, such as sodium glycocholate, have an inhibitory effect on peptidases through ionic and hydrophobic interactions. This confers protection to the peptides and proteins, such as insulin from proteolytic degradation when the bile salts are co-administered orally [58].

Studies investigating the effect of utilizing bile salts as a strategy to enhance drug permeability are summarized in Table 2. Berberine chloride was one of the drugs which had shown an increase in the area under the curve (AUC) through co-administration with sodium deoxycholate in vivo. The investigator discovered that the co-administration of berberine chloride with sodium deoxycholate resulted in a 35.2-fold increase in the plasma concentration of the drug when compared to the control group that received berberine in the absence of the bile salts [70]. In another study, a microcapsule containing taurocholic acid was shown to assist the absorption of gliclazide in diabetic Wistar rats through a targeted release mechanism at pH 7.8 at which the gliclazide solubility improved from approximately 10 µg/mL (without taurocholic acid) to 40 µg/mL after 3 h post-administration, thus producing a better hypoglycemic effect [71]. 

Another derivative of bile salts, sodium taurocholate (10 and 20 mmol/L), increased the permeability of cefquinome from 0.26 ± 0.04 µg/mL to 0.57 ± 0.03 µg/mL and 0.78 ± 0.07 µg/mL, respectively, when evaluated in vivo in rats. It was concluded that the enhancement of the drug absorption was through the modulation of the tight junctions, leading to enhanced paracellular transport [72]. Meanwhile, Moghimipour et al. reported that sodium glycocholate was a better absorption enhancer than sodium taurodeoxycholate, although not statistically significant, for the molecule 5(6)-carboxyfluorescein. In the presence of sodium glycocholate, 12 µg/mL of the dye was able to permeate across the Caco-2 cell lines, while only 10 µg/mL of the compound was able to permeate when co-administered with sodium taurodeoxycholate [73]. Taurodeoxycholic acid (TDCA) is a secondary bile acid, a product of the primary bile acid metabolism in the intestine. A recent study showed that TDCA effectively increased the permeation of EGFR2R-lytic hybrid peptide (epidermal growth factor receptor-binding peptide conjugated with lytic peptide) by the formation of a peptide/bile acid complex, compared to the administration of the peptide alone. The formation of the complex facilitated the widening of the tight junctions along the intestinal epithelium that led to enhanced intestinal absorption [74]. The mechanism for the enhancement in intestinal absorption was evidenced by the ability of the peptide to reduce TEER across Caco-2 monolayers when evaluated in vitro.

**Table 2 pharmaceuticals-15-00975-t002:** Summary of studies investigating the effect of bile salts on the intestinal permeability and oral pharmacokinetic parameters of drugs.

Drug (s)	Absorption Enhancer	Model	Results	Ref.
5(6)-carboxyfluorescein	Sodium glycocholate (SGC) and sodium taurodeoxycholate (STDC)	In vitro: Caco-2 cell	SGC was a slightly better absorption enhancer for the 5(6)-carboxyfluorescein than STDC but not significant (*p* > 0.05).	[73]
Cefquinome	Sodium taurocholate	In vitro: Caco-2 cell	At 2 mmol/L sodium taurocholate, the transportation of cefquinome substantially increased.	[72]
		In vivo: rat intestine	At 10 and 20 mmol/L sodium taurocholate, the absorption of the drug increased in a concentration-dependent manner.	
Berberine chloride	Sodium deoxycholate	In vivo: rat intestine	AUC_0–36h_: 35.3-fold increase	[70]
Gliclazide	Taurocholic acid	In vivo: rat intestine	The microcapsules containing taurocholic acid increased the gliclazide absorption (*p* < 0.01).	[71]
EGFR2R-lytic hybrid peptide	Sodium taurodeoxycholate	In vitro: Caco-2 cell	P_app_: 5.0-fold increase	[74]

### 3.3. Chitosan 

Chitosan is a hydrophilic polysaccharide derived from chitin via N-deacetylation [75]. This biopolymer consists of N-acetyl-d-glucosamine and β-(1–4)-linked d-glucosamine. Due to its biodegradable and biocompatible characteristics, it is extensively used as an excipient either as a tablet disintegrant, release modifier, or even just as a filler. The characteristic of chitosan which is identified to be crucial in exhibiting its permeation-enhancing effect is its mucoadhesive properties. This property is attributed to its positive charge that interacts with the negatively charged glycocalyx of the microvilli along the small intestines [76], resulting in the redistribution of F-actin in the cytoskeleton and the translocation of the tight junctions’ components, ZO-1, and the occludin proteins. This further enables the widening of the tight junctions for the paracellular transport of solutes across the gastrointestinal tract [77,78]. 

The mucoadhesive properties help the drug to adhere to the mucosal surface of the gastrointestinal tract for a prolonged period of time [68]. This increases the residence time of the drugs in the small intestine, which in turn leads to greater drug absorption. Indeed, the permeation-enhancing effect is more pronounced when the grade of chitosan used is of a higher molecular weight, which leads to enhanced permeation rates. The permeation rate and P_app_ value of salvianolic acid B were found to be at a maximum when co-delivered with chitosan displaying a molecular weight of 100 kDa. However, with a decreasing molecular weight, the researchers also observed a decline in the permeation rate of salvianolic acid B [79]. It was proposed that the chitosan, which exhibits a positive surface charge, forms ionic interactions with the negatively charged glycocalyx groups, resulting in the reversibility of the opening tight junctions, which could improve the paracellular drug transport. Another factor that affects the permeation-enhancing effect of chitosan is the degree of deacetylation. Should the grade of chitosan possess a degree of deacetylation that is greater than 80%, the biopolymer was found to exhibit a greater degree of muco-adhesion [80]. It was also discovered that chitosan with a high degree of deacetylation displayed good permeation-enhancing properties at both low and high molecular weights. In contrast, chitosan with a low degree of deacetylation was only effective in enhancing the absorption of the drug molecules across the gastrointestinal tract at higher molecular weights [81]. 

The studies on the role of chitosan as a permeation enhancer are summarized in Table 3. Indeed, some of these results contradict one another; this may be attributed to the different types of intestinal permeability models, the physiological state of the intestinal tissue, and the species used. In addition, the grade of chitosan used may also contribute to such discrepancies in the results. Due to chitosan having an overall pKa 6.5, its absorption-enhancing effects are typically apparent at a pH below 6.5, during which the biopolymer is in a protonated state. This characteristic implies that its absorption-enhancing effects can only be exerted in a limited area along the gastrointestinal tract, as the pH along the intestines may fluctuate to a pH above 6.5 in some parts [82]. Acyclovir was one of the compounds that did not show an improvement in its permeation following co-administration with chitosan. This was due to the interaction between the chitosan and the mucus layers on the intestinal membrane, which altered the reactivity of the chitosan. This caused the biopolymer to have a minimal impact on the Caco-2 monolayer integrity, leading to no enhancement in the absorption of acyclovir [83]. This was further corroborated by a recent study that concluded that the absorption-enhancing effect of chitosan was drug-dependent, with acyclovir displaying unfavorable enhancement of absorption in the presence of chitosan [84]. On the other hand, there are studies that highlighted drugs that displayed an enhanced absorption with the aid of chitosan, and these include molecules such as buserelin, [^14^C] mannitol, and dextran [79,85]. 

The role of chitosan as an absorption enhancer in its salt form was also investigated. Chitosan hydrochloride did not exhibit the same permeability-enhancing effect as the non-salt variant of chitosan. Interestingly, the salt form of chitosan resulted in a decreased C_max_ and AUC_0–36h_ of berberine [86]. This result was attributed to the change in the berberine solubility in the presence of chloride ions, when the drug was co-administered with chitosan hydrochloride. This was explained by the common ion effect in which the chloride ion pairs up and reduces the overall net charge of the berberine. In contrast, trimethyl chitosan was reported as a good absorption enhancer, due to its increased ability to transiently open the tight junctions compared to normal unmodified chitosan [87]. Indeed, it can be seen that chitosan does, to some extent, display good permeation-enhancing properties. Nevertheless, this permeation-enhancing effect is dependent on multiple factors, such as the weight and charge of the biopolymer. A considerable body of work has made strides in understanding the permeation-enhancing effect at a molecular and cellular level. Nevertheless, it can be seen that, indeed, further work is still pivotal to fully understanding and appreciating the mechanisms behind the permeation-enhancing effect of chitosan before this biopolymer can utilized extensively as an excipient for oral drug delivery.

**Table 3 pharmaceuticals-15-00975-t003:** Summary of studies investigating the effect of chitosan and its derivatives on the intestinal permeability and oral pharmacokinetic parameters of drugs.

Drug (s)	Absorption Enhancer	Model	Results	Ref.
Acyclovir	Chitosan	In vitro: Caco-2 cell	P_app_: 124- and 143-fold increase	[83]
		In vivo: rat intestine	AUC_0–12_ and AUC_0–∞_: 0.70- and 0.74-fold decrease C_max_: 0.56- and 0.63-fold decrease T_max_: 1.25- and 1.50-fold increase	
		In vitro: Ussing chamber	P_app_: 1.08- and 2.33-fold increase	
Glucosamine hydrochloride	Chitosan	In vitro: Caco-2 cell	P_app_: 1.9, 2.5 and 4.0-fold increase	[88]
		In vivo: rat intestine	C_max_: 2.8-fold increase T_max_: no change AUC_0−∞_: 2.5-fold increase	
Salvianolic acid B	Chitosan	In vitro: Caco-2 cell	P_app_: 4.43-fold increase	[79]
		In vivo: rat intestine	AUC_0–∞_: 4.25-fold increase	
Berberine	Chitosan hydrochloride	In vivo: rat intestine	AUC_0–36_: no improvement C_max_: no improvement	[86]
	Chitosan	In vivo: rat intestine	AUC_0–36_: maximum 2.5-fold increase	
Amphotericin B	Trimethyl chitosan	In vitro: Caco-2 cell	P_app_: 1.11-fold increase	[87]

## 4. Formulation Strategies to Improve Pharmacokinetics Profile

### 4.1. Solid Lipid Nanoparticles (SLN)

Solid lipid nanoparticles (SLN) have garnered great interest in their ability to improve the oral absorption of poorly soluble drugs. They are a colloidal system derived from a matrix of lipids that retains their solid state at a temperature below 140 °C with sizes ranging from 50 to 1000 nm and able to disperse in an aqueous medium with the aid of surfactants [13,89]. 

Some of the advantages of employing SLN as a drug delivery system for oral administration include mitigating the degradation of entrapped drugs, as well as providing some control over the rate of drug release. Another advantage of utilizing this formulation approach is the ability to avoid hepatic first-pass metabolism through the intestinal lymphatic uptake. This is because the lipophilic nature of the SLN causes the nanocarrier to drain into the thoracic lymph before entering the systemic circulation near the left subclavian vein, thus circumventing the hepatic first-pass metabolism [90,91]. One of the reasons for the absorption-enhancing ability conferred by SLN is attributed to the interaction of the P-gp efflux pump, which causes the substrate to be unavailable for transport. This was confirmed by Garg et al. (2016), who discovered that by formulating lumefantrine as the SLN, they were able to enhance the oral bioavailability of the drug, which is a substrate for P-gp, by 2.7-fold relative to when the lumefantrine was delivered in 0.25% *w*/*v* Na CMC suspension by oral gavage. The SLN used consisted of a binary lipid mixture of stearic acid and caprylic acid stabilized with the non-ionic surfactant, D-alpha tocopheryl polyethylene glycol 1000 succinate (TPGS) and formulated Poloxamer 188 [92]. Another advantage of formulating poorly soluble drugs by using SLN is the ability of the nanocarrier to increase the dissolution rate of the drug within the gastric lumina, thus generating a concentration gradient to promote absorption across the gastrointestinal membrane. Such an enhanced dissolution rate is attributed to the submicron size of the SLN, which provide a large surface area for drug dissolution [92]. 

Valdes et al. reported that the bioavailability of 4-(N)-docosahexaenoyl 2′, 2′-difluorodeoxycytidine (DHA-dFdC) improved from 113.55 μg•h/mL to 143.44 μg•h/mL when delivered as a SLN formulation. A conclusion on the exact absorption-enhancing mechanism was not fully elucidated, yet the researchers have proposed a few hypotheses for the absorption-enhancing effect observed. Firstly, the SLN may be absorbed through the microfold cells in the Peyer’s patches, leading to the migration of the nanocarrier into the lymphatic system thus obviating the first-pass metabolism and leading to enhanced bioavailability. Secondly, the authors also proposed that the SLN release lipids upon digestion that alter the gastrointestinal fluid media, that then lead to the enhanced solubility and dissolution of the DHA-dFdC, leading to improved absorption. In addition, it was postulated that the SLN provided a barrier that prevented the drug from being susceptible to enzymatic degradation along the gastrointestinal tract compared to the DHA-dFdC alone, thus enabling the absorption of the intact drug across the intestinal walls [85]. The enhanced stability of the delivered payload when formulated as a SLN was supported by the findings of Ansari et al., that showed that approximately 61.6% and 92.9% of insulin in a SLN formulation remained at 1 h when incubated with pepsin and trypsin, respectively, in comparison to the free insulin solution (45% in pepsin and 53% in trypsin) [93]. These results demonstrated the capability of SLNs in conferring protection on the delivered payload during transit and absorption through the gastrointestinal tract.

Other than that, SLN can improve the uptake of drugs via the intestinal lymphatic systems. Examples of drugs that displayed improved intestinal lymphatic uptake via the use of SLN include curcumin and asenapine maleate [90,94]. N-carboxymethyl chitosan-coated curcumin-loaded SLN (NCC-SLN) demonstrated a 6.3-fold increase in the lymphatic uptake relative to the curcumin solution, which circumvents the hepatic first-pass metabolism as the drug is delivered directly into the systemic circulation from intestinal lymphatics, and subsequently improves the bioavailability of the drug by 9.5-fold compared to the curcumin solution [90]. Patel et al. reported a greater increase in the bioavailability of asenapine maleate by 50.19-fold when the drug was administered as SLN orally in female Sprague Dawley rats, relative to the asenapine maleate dispersion [94]. Such an enhancement was attributed to the fact that the SLN promoted the asenapine maleate absorption through claveole- and clathrin-mediated endocytosis, which led to an increased uptake into the intestinal lymphatic system. In addition, the author also attributed such an enhancement in oral bioavailability to the role of the surfactant, TPGS, which was able to enhance the chylomicron release due to its role as a lymphotropic excipient. However, the authors found that the SLN took longer to achieve the maximum plasma concentration (t_max_) than the drug dispersion alone. Such an observation may be attributed to the time it takes for the formation of lipoproteins at the rough endoplasmic reticulum followed by core expansion at the smooth endoplasmic reticulum. This process then culminates in the formation of a large lipoprotein called chylomicron which then associates with the drug administered before transiting into the intestinal lymphatics. All of the literature reviewed is summarized in Table 4 below.

### 4.2. Dimers

A dendrimer is a synthetic polymer with a highly branched amidoamine structure and an ethylenediamine core [96]. The repeated branching results in the dendrimer having a hollow interior and a dense exterior surface [97]. It also has a nanosized and spherical structure. Drugs may be bound to the polymer surface or loaded into the central core, depending on the physicochemical properties of the drug. 

Several generations of poly amido amine (PAMAM) dendrimers have been investigated in order to understand their absorption-enhancing effect on the drugs with poor permeability across the intestinal layer. The summary of these studies is shown in Table 5. Yan et al. investigated the effects of an acetylated G2 PAMAM dendrimer on the intestinal absorption of poorly absorbable water-soluble drugs, using an in situ closed-loop method in rats [98]. The acetylated G2 PAMAM was synthesized by reacting with acetic anhydride with G2 PAMAM to produce a primary amine-acetylated G2 PAMAM dendrimer (Ac-G2), in which the primary amine group on the dendrimer surface was converted to acetamide. Among the various acetylation levels, Ac50-G2 displayed the greatest absorption-enhancing effect on the permeation of the fluorescein isothiocyanate-labelled dextrans (FD4), 5(6)-carboxyfluorescein (CF), and alendronate. However, the same result was not observed for the macromolecular drug, FD10. The possible reason for such an observation may be attributed to the effect of the PAMAM dendrimers on loosening the tight junctions along the gastrointestinal tract. Although the dendrimer may loosen the tight junctions and enable the permeation of fluorescein isothiocyanate-labelled dextrans (FD4), 5(6)-carboxyfluorescein (CF), and alendronate, the size of the pores formed were insufficient to allow the paracellular transport of the macromolecular drug, FD10. It was also discovered that different generations of dendrimers had differing effects on the absorption-enhancing effects, with G2- and G3-acetylated PAMAM dendrimers being more effective than G0- and G1-acetylated PAMAM dendrimers. Yan and co-workers also highlighted that the dendrimer was shown to be safe and did not result in any observable damage to the intestinal lining following intestinal administration, when the concentrations of acetylated PAMAM used were below 0.50 w/w. 

A study by Qi et al. (2015) supported tight junction modulation as one of the absorption-enhancing mechanisms of dendrimers. The investigators discovered that, under their experimental conditions, simvastatin showed an improved water solubility as well as increased oral bioavailability when the drug was delivered as a molecular complex with PAMAM (G5-NH_2_). The increase in the absorption of simvastatin when delivered as a complex with PAMAM (G5-NH_2_) is attributed to the interruption of the occludin-1, which further enlarged the tight junctions to allow a greater entry of the simvastatin-PAMAM complexes across the gut epithelial. This enhancement in absorption is further promoted by the enhanced solubilization of simvastatin coupled with the inhibition of P-gp by the dendrimer [99]. However, Sadekar et al. suggest that this mechanism may not be prominent in enhancing the oral absorption in mice, to which they deduced that the variables in present within in vivo gastrointestinal system such as intestinal fluid dilution, gastrointestinal transit and mucosal barrier might reduce the effective concentration of the dendrimer and incubation time, which may result in the dendrimer being ineffective in modulating the tight junctions [100]. 

The surface charge of a dendrimer may also play a role in influencing the efficiency of the dendrimer as a permeation enhancer. Sadekar et al. discovered that PAMAM G4.0, which is a cationic dendrimer, and G3.5, which is an anionic dendrimer, were equally effective in enhancing the absorption of camptothecin in vivo, using a female CD-1 mice model. Both caused an approximate two- to three-fold oral absorption enhancement of camptothecin in vivo at 2 h, as compared to camptothecin alone. They attributed the findings to several reasons, which are an increased transcellular uptake of the PAMAM-associated camptothecin via the endocytic mechanisms or the enhanced solubilization of PAMAM-associated camptothecin in simulated gastric fluid. However, the researchers discovered that the drug was associated more with the cationic dendrimer, PAMAM G4.0 (80%), than with its anionic counterpart, PAMAM G3.5 (20–30%). This was attributed to the drug being negatively charged, at the pH at which the formulation was prepared, leading to strong electrostatic interaction with the cationic surface of the G4.0 PAMAM dendrimer [100]. 

Besides that, the charge of the PAMAM dendrimers may play a pivotal role in the adsorptive capacity of the dendrimer with the gastrointestinal lining, due to the interaction of its positive charge of the G4.0 dendrimer surface with negative-charged components of the intestine membrane. This may lead to the adhesion of the gastrointestinal wall, thus promoting the residence time of the drug along the gastrointestinal lining leading to enhanced intestinal permeability of anionic drugs. Another study by D’Emanuele et al. emphasized propranolol-lauroyl-G3 dendrimer conjugates’ capacity to enhance the propranolol permeation by approximately 3.5 times that of propranolol alone, through the inhibition of the P-gp efflux and endocytosis-mediated transepithelial transport [101]. 

In addition, the PAMAM dendrimers have a size-dependent solubilization-enhancing effect, where the higher generation PAMAM (G4 PAMAM dendrimer) exhibits a greater drug solubility than those with a lower generation (G2 and G3 PAMAM dendrimers) at the same concentration, as shown in the study by Yiyun and Tongwen. This is related to the increasing number of primary and tertiary amines within the dendrimer, with the larger generation size permitting a higher entrapment of more hydrophobic molecules [102]. 

**Table 5 pharmaceuticals-15-00975-t005:** Summary of studies investigating the effect of dendrimers on the intestinal permeability and oral pharmacokinetic parameters of drugs.

Drug (s)	Model	Results	Ref.
5(6)-carboxyfluorescein (CF), fluorescein isothiocyanate-labeled dextrans (FD4, FD10) and alendronate	In vitro: diffusion chamber	P_app_: increased except for FD10.	[98]
	In vivo: rat intestine	The greatest AUC achieved in the presence of Ac50-G2 (0.5%, *w*/*v*).	
Camptothecin	In vivo: rat intestine	AUC: 2- to 3-fold increase C_max_: increased T_max_: no change	[100]
Simvastatin	In vivo: rat intestine	AUC: increased C_max_: increased T_max_: 1.5-fold increase	[99]
	In vitro: Caco-2 cell	P_app_: increased	
Propranolol	In vitro Release Study (dialysis sac)	P_app_: increased	[102]
	In vitro: Caco-2 cell	AUC: increased	[101]

Indeed, from the studies highlighted, it is apparent that the PAMAM dendrimer has a huge potential in enhancing the solubilization and absorption of poorly soluble drugs. Such an enhancement is attributed to several mechanisms, as highlighted above. As newer generations of PAMAM dendrimers emerge, which are less toxic yet still as effective as the current generations of the dendrimer, the utilization of the PAMAM dendrimer will surely gain popularity as a nanocarrier for the delivery of drugs through the oral route.

### 4.3. Nanoemulsions

Nanoemulsions are translucent or transparent water-in-oil (w/o) or oil-in-water (o/w) droplets [103] that are thermodynamically stable nanoformulations prepared from a mixture of water, oil, and surfactants, co-surfactants in an aqueous phase with droplet sizes ranging 1 to 200 nm [14,103,104,105]. The addition of a co-surfactant improves the emulsion stability by increasing the fluidity, as well as by a disordering effect on the surfactant film [106] that increases the drug loading and the formation of an extemporaneous nanoemulsion [107,108]. The diameter of the oil droplets is usually in the range 50–200 nm, as compared to conventional emulsions which have the size range from 1 to 100 µM [109]. The use of a pseudo-ternary phase diagram aids in the identification of the optimum composition of water, oil, and surfactant that results in the formation of a nanoemulsion region via the titration method. Nanoemulsions are widely used due to their advantages that include the protection of therapeutics against chemical and enzymatic degradation, high solubilization capacity, improved drug absorption [110], rapid onset of C_max_, ease of fabrication, as well as the facile scale-up of the manufacturing process [111,112,113]. In addition, the enhanced colloidal stability conferred by a nanoemulsion mitigates the propensity of the formulation to coalesce and flocculate over a long storage period [114].

A number of studies have investigated the viability of utilizing nanoemulsion in facilitating the transport of various drugs across the gut epithelium. The summary of these studies is shown in Table 6. Gao et al. (2011) designed a candesartan cilexetil-loaded nanoemulsion (CCN), which was composed of candesartan–cilexetil, soybean oil, Solutol HS-15, and Tween 80 (1:6:10:20, w/w). The investigators found that there was a significant increase in the CCN permeability by 1.75-fold in the duodenum, 1.93-fold (jejunum), and 1.84-fold (ileum) using an in situ single-pass intestinal perfusion model relative to the conventional candesartan–cilexetil suspension [111]. The group conducted a pharmacokinetic study that highlighted the differences between the nanoemulsion and suspension, which reported a 27-fold increase in the C_max_ of candesartan upon oral administration of the CCN compared to the free candesartan–cilexetil suspension at 0.59 h. The underlying mechanism for the reported 27-fold increase in the C_max_ of the candesartan was due to the ability of the nanoemulsion to be readily internalized by the enterocytes via clathrin-mediated endocytosis relative to the candesartan–cilexetil suspension. This then leads to the enhanced formation of intestinal chylomicron within the enterocytes for subsequent lymphatic transport into the systemic circulation. In addition, the nanosize droplets of the emulsion also provide a high surface area to volume ratio for the droplets to interact with the apical membrane of the enterocytes, thus increasing the uptake of the nanodroplets by the enterocytes [113].

Li et al. (2015) investigated the effectiveness of the nanoemulsion as a nanocarrier for the delivery of curcumin. The group formulated curcumin into a nanoemulsion that was stabilized by whey protein isolate as an emulsifier. The nanoemulsion displayed an average particle size of 208 nm and enhanced the solubility and stability of curcumin against pepsin-induced proteolytic degradation while resulting in enhanced permeation across the Caco-2 cell monolayers by two-fold relative to the unformulated curcumin. The authors attributed the enhanced permeation across the cell monolayers as being due to two possible mechanisms [109]. The first one was through the digestion–absorption pathway, which was regarded to be the major permeation mechanism. The digestion of the nanoemulsion by trypsin and lipase led to the release of curcumin, allowing it to be absorbed across the intestinal membrane. This is because the nanoemulsion was found to be sensitive to trypsin but not to pepsin. The other one was the direct diffusion pathway, that does not involve digestion. The nanosized droplets made it easier for the system to diffuse directly across the small intestinal layer. In the same study, the investigators compared the permeation-enhancing properties between the curcumin nanoemulsion and the β-lactoglobulin–curcumin complex. The group discovered that the nanoemulsion had slightly better permeability of curcumin (7.07 × 10^−5^ cm/s), albeit not significant, as compared to the complex (7.02 × 10^−5^ cm/s). 

Recent studies have established that the inhibition of P-gp-mediated efflux transportation as the latest mechanism responsible for the absorption enhancement effect of nanoemulsions. Berberine hydrochloride is a P-gp substrate and suffers poor oral absorption, due to the efflux of the drug upon entering the enterocytes. The incorporation of the drug into a nanoemulsion has been an effective and successful strategy to reduce the efflux of berberine by the transporter, thereby enhancing the drug absorption, as shown by a 4.4-fold higher relative oral bioavailability than the delivery of the drug using a conventional drug suspension [114]. This was further supported by a study by Chen et al. (2018) that showed that the P_app_ and K_a_ of paeonol nanoemulsion increased significantly by 0.65- and 1.64-fold in the colon, respectively, relative to the paeonol suspension [115]. This study illustrated that, although the colon has the largest expression of P-gp, the use of nanoemulsions has been shown as a viable strategy to mitigate the excretion of P-gp substrates. 

The greater surface area of nanoemulsions relative to microemulsions and the ability to modify the tight junctions are other reported advantages of nanoemulsions. Recently, Anuar and co-workers demonstrated that the absorption-enhancement effect of an ibuprofen-loaded nanoemulsion relative to conventional ibuprofen oil-in-water emulsion (control), and microemulsion. In this study, sucrose esters oleate was used as a surfactant to form an ibuprofen microemulsion and nanoemulsion. The author also showed that the solubility of ibuprofen was enhanced when the drug was formulated as a nanoemulsion. Such enhancement in solubility enabled the drug to diffuse across the unstirred water layer in the intestine and into the enterocytes, leading to enhanced oral bioavailability in vivo [110]. It can be seen from these studies that formulating drug molecules into a nanoemulsion confers several advantages in enhancing the oral bioavailability while providing protection to the drug as the molecule is transported within the gastrointestinal lumina. Nevertheless, meticulous and judicious selection of the excipients along with the careful identification of the right composition needed to form the nanoemulsion phase are pertinent in the development of such systems. However, as the range of commercially available nanoemulsions continues to increase (e.g., Liple^®^, Cleviprex, Limethasone, Vitalipid^®^, and Ropion), it can be deduced that nanoemulsions have a strong clinical translational value and may be a commercially viable and effective strategy to enhance the oral bioavailability of poorly absorbed drugs [116].

## 5. Conclusions

Chitosan, surfactants, and bile salts are among the absorption enhancers that have been recognized for their absorption-enhancing effect. In addition, nano carrier-based drug delivery systems, such as solid lipid nanocarriers, nanoemulsions, and dendrimers, have been proven to have noteworthy contributions to the increased intestinal permeability. More significant progress in the nanocarrier-based drug delivery systems is expected to be discovered and developed in the future. Nonetheless, the effectiveness of this drug delivery system in humans requires more extensive clinical study, as the in vitro and in vivo studies on animals do not reflect the exact state of the human intestinal tract, due to interspecies variations. Nevertheless, these pre-clinical studies have given insight into the value, the potential mechanisms as well as some insight on the safety of these permeation enhancers. It is expected that through judicious choice of permeation enhancers, coupled with the meticulous evaluation of these formulations, such a drug delivery strategy would soon be translated and become more commonplace in a clinical setting. This is of great therapeutic value, as such formulation strategies are capable of enhancing the oral bioavailability for the poorly soluble and poorly permeable drugs, without the need of administrating such therapeutics through a more invasive manner, such as hypodermic injection or intravenous administration.

## Figures and Tables

**Figure 1 pharmaceuticals-15-00975-f001:**
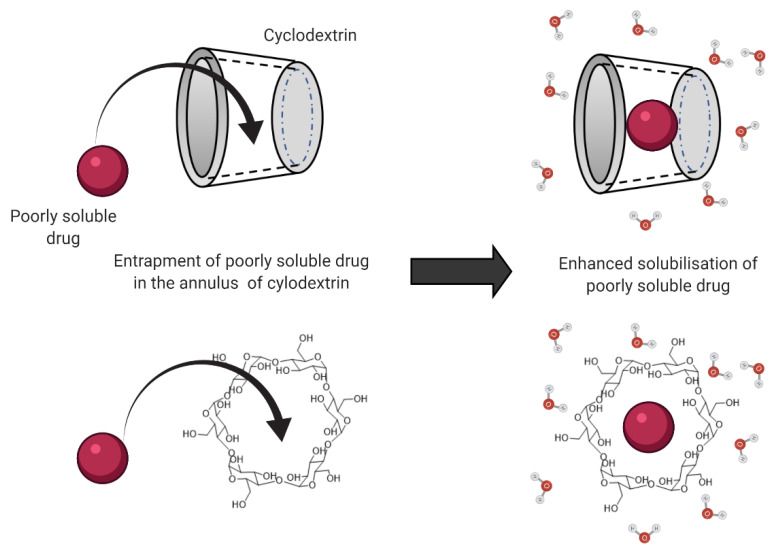
Mechanism for cyclodextrin-enhanced solubility of poorly soluble drugs through entrapment and complex formation with the drug molecule.

**Table 1 pharmaceuticals-15-00975-t001:** Summary of studies investigating the effect of solubilizing agents on the intestinal permeability and oral pharmacokinetic parameters of drugs.

Drug (s)	Absorption Enhancer	Model	Results	Ref.
[^14^C]-mannitol	Sucrose laurate	In vitro: Caco-2 cell	P_app_: 9-fold increase	[57]
	Sucrose laurate	In vitro: Ussing chamber	P_app_: 2.6-fold increase	
Insulin	Sucrose laurate	In situ: rat jejunum and colon	Relative bioavailability (F, %): 8.9% increase	[57]
Fluorescein, atenolol, rhodamine 123, and vinblastine	Sucrose laurate	In vitro: Caco-2 cell	P_app_: several folds increase for all drugs.	[65]
Carbamazepine	Cyclodextrins	In vivo: dogs	T_max_: 0.6-fold decrease C_max_: 0.004-fold increase	[63]
Erlotinib	Cyclodextrins	In vivo: rats	T_max_: 5.4-fold decrease C_max_: 3.2-fold increase AUC: 3.6-fold increase	[59]

**Table 4 pharmaceuticals-15-00975-t004:** Summary of studies investigating the effect of solid lipid nanoparticles on the intestinal permeability and oral pharmacokinetic parameters of drugs.

Drug (s)	Model	Results	Ref.
Lumefantrine	In situ: single pass intestinal permeability study	Cellular uptake: 3-fold increase K_a_: 2.96-fold increase	[92]
	In vivo: rat intestine	AUC and C_max_: 2.7-fold increase T_max_: no change	
Curcumin	In vivo: rat intestine	Lymphatic uptake: 6.3-fold increase Oral bioavailability: 9.5-fold increase C_max_: several folds increase T_max_: 2-fold increase AUC: increased	[90]
Asenapine maleate	In vitro: Caco-2 cell	P_app_: increased	
	In vivo: rat intestine	Bioavailability: 50.19-fold increase AUC: increased C_max_: 20.78-fold increase T_max_: 8-fold increase	[94]
4-(N)-docosahexaenoyl 2′, 2′-difluorodeoxycytidine (DHA-dFdC)	In vitro: simulated gastrointestinal fluids	C_max_: increased T_max_: decreased AUC: increased	[95]
Insulin	Ex vivo: rat everted intestinal sac	P_app_: 2-fold increase C_max_: increased AUC: increased	[93]

**Table 6 pharmaceuticals-15-00975-t006:** Summary of studies investigating the effect of nano-emulsions on the intestinal permeability and oral pharmacokinetic parameters of drugs.

Drug (s)	Model	Results	Ref.
Paeonol	In situ: single-pass intestine perfusion	P_app_: 1.64-fold increase K_a_: 0.65-fold increase	[115]
	In vitro: everted gut sacs	P_app_: increased (*p* < 0.01)	
	In vitro: Caco-2 cell	P_app_: increased	
	In vivo: rat intestinal uptake	AUC_0→t_: 4.27-fold increase C_max_: 4.02-fold increase T_max_: 40-min increase	
Berberine hydrochloride	In vivo: rat intestinal uptake	AUC: 4.4-fold increase C_max_: 1.6-fold increase T_max_: 4.3-fold increase	[114]
	In vitro: Caco-2 cell	P_app_: increased to 0.574 ± 0.18 × 10^−8^ cm/s	
Curcumin	In vitro: Caco-2 cell	The digested nanoemulsion had the highest permeation rate (7.07 × 10^5^ cm/s)	[109]
Candesartan cilexetil	In situ single-pass intestinal perfusion	Cellular uptake: 1.75-, 1.93-, and 1.84-fold increase in the duodenum, jejunum, and ileum, respectively.	[111]
	In vitro: Caco-2 cell	The cellular uptake of CCN at 4 °C reduced 92% compared with that at 37 °C (*p* < 0.01)	
	In vivo: rat intestinal uptake	AUC: 10-fold increase C_max_: 27-fold increase T_max_: no change	
Ibuprofen	In vitro diffusion chamber: rat intestinal membrane	P_app_: 10.6-fold	[110]
	In vivo: rat intestinal uptake	AUC _0–6h_: 2.2-fold increase C_max_: 27-fold increase T_max_: no change	

## Data Availability

Data sharing not applicable.

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
