# Peer review of "Intestinal Absorption Study: Challenges and Absorption Enhancement Strategies in Improving Oral Drug Delivery"

_pharmaceuticals, 2022, doi:10.3390/ph15080975_

Round 1
Reviewer 1 Report
this review is definitely well written, English language is good and the article is well contructed. The part regarding ways to improve absorption is good, only the first part on absorption limitation causes should be improved.
In the introduction part, the authors described a well-known principle, the BCS classification but I suggest to add sentences and discussion about the new classification suggested by Benet, the BDDCS. Indeed, this new classification is directly in the field of the absorption enhancement strategies as finally, most of the time bioavailibity is limited by metabolism.
In the mucous part, add some exemples of drug where mucous facilitate or block absorption to complete the description. In the same way, add exemples of drug absorbed by tight jonctions.
In the transporter part, as this article is a review, precise the actual nomenclature name of the transporter (ABCB1 for P-gp, ABCG2 for BCRP...) and add reference to justify the fact that P-gp has a regional distribution in the gut and precise the opposite distribution of intestinal cytochromes (Mouly et al or Bruyere et al for instance). Discuss the consequences on drug bioavailibity of this link between P-gp and cytochromes.
The auhors should also describe, discuss and add exemple of lymphatic absorption processes before showing galenic forms which pass through lymphatic absorption.
In line 232-233, I am not totally agree with the shortcut made by the authors. Indeed, the first-pass metabolism is not only due to intestine but also (and mostly) by the liver. Rephrase the sentence because it is ambiguous.
In the exemples cited in line 253, the authors shoudl add midazolam as it is metabolized half by intestine and half by the liver and it is a reference drug. Also precise in this part the regional distribution of cytochromes and precise that not only CYP3A4 is present but other cytochromes too. In line 262-263, do you have some exemples on intestinal enzymatic induction?
Line 433-435, add hypothesis to justify this point.
Delete of line 467
Line 493-495, I would not wrtie inhibition of P-gp. Indeed, inhibition suggesting direct or indirect interaction which leads to the non fonction of the transporter. In this case, the P-gp is always active, it's just the substrate which is not available, it could not be considered as an inhibition... Please rephrase this sentence.
Al last, check space before words in the complete article.
Author Response
Reviewer comments:
This review is definitely well written, English language is good and the article is well constructed. The part regarding ways to improve absorption is good, only the first part on absorption limitation causes should be improved.
Reviewer point #1:
In the introduction part, the authors described a well-known principle, the BCS classification but I suggest to add sentences and discussion about the new classification suggested by Benet, the BDDCS. Indeed, this new classification is directly in the field of the absorption enhancement strategies as finally, most of the time bioavailability is limited by metabolism.
Author response #1:
Thank you for the suggestion. We have included the new classification suggested by Benet, the BDDCS in the main text (page 2, lines 62-66).
Reviewer point #2:
In the mucous part, add some exemples of drug where mucous facilitate or block absorption to complete the description. In the same way, add exemples of drug absorbed by tight jonctions.
Author response #2:
Thank you for the input.
We have now included examples of drugs and therapeutics which are affected/blocked by mucus binding along the GI tract in Section 2.1 Mucous of the review. Examples of these drugs and therapeutics include peptides and proteins in the range of 14.4–168 and 3.4–66 kDa [1] as well as small drug molecules such as albuterol, rifampicin, p-amino-salicylic acid, isoniazid, pyrazinamide, pentamidine [2], pivampicillin, isoprenaline (isoproterenol), isoetharine and rimiterol [3] (page 2, lines 131-135).
We have also added examples of drugs and therapeutics by which their absorption occurs through tight junctions in Section 2.2 Tight Junction. Examples include peptide drugs, such as octreotide, vasopressin analog desmopressin and thyrotropin-releasing hormone [4]. In addition, the work by Tam et al have shown that amphoteric drugs such as acyclovir, amdinocilin, ganciclovir, piroxicam and trovafloxacin undergoes extensive paracellular transport via tight junctions [5] (page 4, lines 179-83).
Reviewer point #3:
In the transporter part, as this article is a review, precise the actual nomenclature name of the transporter (ABCB1 for P-gp, ABCG2 for BCRP...) and add reference to justify the fact that P-gp has a regional distribution in the gut and precise the opposite distribution of intestinal cytochromes (Mouly et al or Bruyere et al for instance). Discuss the consequences on drug bioavailibity of this link between P-gp and cytochromes.
Author response #3:
Thank you for the input. We have:
- Added the precise nomenclature for the transporters as requested in Section 2.3 Efflux Transporter (page 4, lines 187-191).
- In addition, we have added references and discussion pertaining P-gp and cytochrome CYP3A4 regional distribution in the gut and along with it consequences on drug bioavailability in Section 2.3 Efflux Transporter (page 5, lines 214-239).
Reviewer point #4:
The auhors should also describe, discuss and add exemple of lymphatic absorption processes before showing galenic forms which pass through lymphatic absorption.
Author response #4:
Thank you for pointing this out. We have included subsection (2.6) of intestinal lymphatic absorption in the text (page 7, lines 340-352).
Reviewer point #5:
In line 232-233, I am not totally agree with the shortcut made by the authors. Indeed, the first-pass metabolism is not only due to intestine but also (and mostly) by the liver. Rephrase the sentence because it is ambiguous.
Author response #5:
Thank you for the suggestion. We agree that the statement is ambiguous and have revised the sentences as suggested (page 6, lines 295-299).
Reviewer point #6:
In the exemples cited in line 253, the authors shoudl add midazolam as it is metabolized half by intestine and half by the liver and it is a reference drug. Also precise in this part the regional distribution of cytochromes and precise that not only CYP3A4 is present but other cytochromes too. In line 262-263, do you have some exemples on intestinal enzymatic induction?
Author response #6:
Thank you for the input. We have added midazolam and a rephrase on the cryptochromes that are involved in the metabolism in the main text (page 7, lines 319-321).
Reviewer point #7:
Line 433-435, add hypothesis to justify this point.
Author response #7:
We have added the hypothesis as in the main text (page 12, 515-517).
Reviewer point #8:
Delete of line 467
Author response #8:
We have now deleted line 467 as suggested.
Reviewer point #9:
Line 493-495, I would not wrtie inhibition of P-gp. Indeed, inhibition suggesting direct or indirect interaction which leads to the non fonction of the transporter. In this case, the P-gp is always active, it's just the substrate which is not available, it could not be considered as an inhibition... Please rephrase this sentence.
Author response point #9:
Thank you for pointing this out. We indeed agree with the reviewer and we have rephrased the sentence as suggested (page 14, line 576-577).
Reviewer point #10:
Al last, check space before words in the complete article.
Author response point #10:
Thank you for pointing this out. We have checked the format and made the necessary changes throughout the manuscript accordingly.
References:
[1] M. Boegh and H. M. Nielsen, “Mucus as a barrier to drug delivery - Understanding and mimicking the barrier properties,” Basic and Clinical Pharmacology and Toxicology. 2015, doi: 10.1111/bcpt.12342.
[2] P. G. Bhat, D. R. Flanagan, and M. D. Donovan, “Drug binding to gastric mucus glycoproteins,” Int. J. Pharm., vol. 134, no. 1, pp. 15–25, 1996, doi: https://doi.org/10.1016/0378-5173(95)04333-0.
[3] C. F. George, “Drug Metabolism by the Gastrointestinal Mucosa,” Clin. Pharmacokinet., vol. 6, no. 4, pp. 259–274, 1981, doi: 10.2165/00003088-198106040-00002.
[4] S. Maiti, Nanometric Biopolymer Devices for Oral Delivery of Macromolecules with Clinical Significance. Elsevier Inc., 2017.
[5] K. Y. Tam, A. Avdeef, O. Tsinman, and N. Sun, “The permeation of amphoteric drugs through artificial membranes - An in combo absorption model based on paracellular and transmembrane permeability,” J. Med. Chem., vol. 53, no. 1, pp. 392–401, 2010, doi: 10.1021/jm901421c.
Reviewer 2 Report
The manuscript describes an important problem of the pharmacokinetics of an indispensable process in the development of a new drug.
The authors could have written some information about the absorption process, e.g. about the achievements of J Wagner.
Author Response
Reviewer point 1#:
The manuscript describes an important problem of the pharmacokinetics of an indispensable process in the development of a new drug.
The authors could have written some information about the absorption process, e.g. about the achievements of J Wagner.
Author response #2
Thank you for the suggestion. We now included some information on the achievements and contributions of JG Wagner who provides fundamental principles on biopharmaceutics which was first published in 1961 and several other publications later that have had an enormous impact on pharmaceutical research. This information is now included in the introduction section (Page 2, Line 67-76)
Reviewer 3 Report
Ms. Ref. No.: pharmaceuticals-1825195-peer-review-v1
Title: Intestinal absorption study: challenges and absorption enhancement strategies in improving oral drug delivery
Authors: Maisarah Azman, Akmal H Sabri, Qonita Kurnia Anjani, Mohd Faiz Mustaffa and Khuriah Abdul Hamid
Overview and general recommendation:
The review aims to provide an overview of the challenges faced by oral drug delivery and selected strategies to enhance oral drug absorption, including the application of absorption enhancers and nanocarrier-based formulations based on in vitro, in vivo, and in situ studies. Overall, I found the review is well designed and written. However, I ask that the authors specifically address the following minor recommendations before publication.
Comments to the authors:
1. I think the manuscript needs revision for grammar and spelling e.g., in line 586, correct “these mechanism”.
2. In 4.3. Nanoemulsions section, the definition needs revision because there is no mention of cosurfactant which plays an important role in the formulation of nanoemulsion. In addition, the size range is between 1 and 100 μM, I think this unit is incorrect. Please revise.
3. In the conclusion section, I recommend the use of the first two sentences in the introduction section. Also, it is not recommended to cite references in the conclusion.
Author Response
The review aims to provide an overview of the challenges faced by oral drug delivery and selected strategies to enhance oral drug absorption, including the application of absorption enhancers and nanocarrier-based formulations based on in vitro, in vivo, and in situ studies. Overall, I found the review is well designed and written. However, I ask that the authors specifically address the following minor recommendations before publication.
Comments to the authors:
Reviewer 3 point #1:
I think the manuscript needs revision for grammar and spelling e.g., in line 586, correct “these mechanism”.
Author response point #1:
We have now improved the grammar and spelling errors found in this manuscript and we have corrected “these mechanism” to “this mechanism” (Page 16, Line 669).
Reviewer 3 point #2:
In 4.3. Nanoemulsions section, the definition needs revision because there is no mention of cosurfactant which plays an important role in the formulation of nanoemulsion. In addition, the size range is between 1 and 100 μM, I think this unit is incorrect. Please revise.
Author response point #2:
Thank you for the suggestion, we have now revised the definition of nanoemulsion to include the important role of cosurfactant in the formulation. We have also rephrased the sentence about the size range between 1 and 100 μM which was referring to conventional emulsions to make it clearer to the readers. (Page 18, Line 712 – Line 719)
Reviewer 3 point #3:
In the conclusion section, I recommend the use of the first two sentences in the introduction section. Also, it is not recommended to cite references in the conclusion.
Author response point #3
Thank you for the suggestion, we have now moved the first two sentences to the Introduction section (Page 1, Line 35 – Line 43) and removed reference citation in the conclusion.